# Key Proteins and Metabolic Pathways Involved in 24-Epibrasionlide Improving Drought Tolerance of *Rhododendron delavayi* Franch

**Yan-Fei Cai** [1,2,†] , **Lu Zhang** [1,2,†] , **Lv-Chun Peng** [1,2] , **Shi-Feng Li** [1,2] , **Jie Song** [1,2] , **Wei-Jia Xie** [1,2] and **Ji-Hua Wang** [1,2,*]

1    Flower Research Institute of Yunnan Academy of Agricultural Sciences, Kunming 650205, China; caiyanfei1013@126.com (Y.-F.C.); changjiangyulu@163.com (L.Z.); green315@126.com (L.-C.P.); kylin7023@126.com (S.-F.L.); songjie2591@126.com (J.S.); weijiax@163.com (W.-J.X.)
2    National Engineering Research Center for Ornamental Horticulture, Kunming 650200, China
*    Correspondence: wjh0505@gmail.com; Tel.: +86-138-8882-2307
†    These two authors contributed equally to this work.

**Abstract:** *Rhododendron delavayi* Franch. is a famous ornamental plant. However, seasonal drought caused by a monsoon climate seriously limits its growth and flowering performance in urban gardens. Our previous study has found that brassinosteroids (BRs) can improve the tolerance of *R. delavayi* to drought stress. Here, we employed a data-independent acquisition (DIA) approach to compare the protein expression profiles under drought treatment (D) and pre-treatment with BR before drought treatment (BR). With an increase in drought stress, the net photosynthetic rate, stomatal conductance, and transpiration rate in the BR treatment showed more stable changes that were significantly higher than those in the D treatment. However, the contents of malondialdehyde, soluble sugar, soluble protein, and the activity of superoxide dismutase (SOD), peroxidase, and catalase showed opposite trends. The pre-treatment with BR alleviated the negative effect of drought stress on the photosynthetic performance of *R. delavayi*. A total of 3453 differentially expressed proteins (DEPs) were identified, and 683 DEPs were significantly expressed in the D and BR treatments. The DEPs uniquely expressed in the BR treatment participated in the pathways of "ribosome", "ether lipid metabolism", "photosynthesis", and "oxidative phosphorylation". The improvement effect of the BR treatment on the drought tolerance of *R. delavayi* was mainly attributed to improved photosynthesis by alleviating stomatal closure and oxidative stress, maintaining the integrity and stability of the ribosomal complex to mediate protein synthesis and the balance between energy metabolism and carbon metabolism. Our study presents a comprehensive understanding of the key proteins and metabolic pathways related to the response of *R. delavayi* to drought and will contribute to the breeding of drought-tolerant rhododendrons.

**Keywords:** brassinosteroids; data-independent acquisition; drought stress; proteomic analysis; *Rhododendron delavayi*

## 1. Introduction

Drought stress is one of the main environmental factors limiting plant growth and productivity [1,2]. However, plants have developed a series of physiological and anatomical modifications in response to drought stress [3]. Stomatal closure, which is generally considered to be the initial and most rapid response of plants to drought, can reduce plant water loss and improve water use efficiency. However, stomatal closure may decrease the photosynthetic performance under drought stress, therefore, reducing carbon assimilation [4]. In addition, drought stress can destroy the membrane structure of plants and cause oxidative stress [3]. Various small molecular compounds, such as proline, soluble sugars, and soluble protein, can be synthesized and accumulated to maintain plant water

uptake and membrane integrity under drought stress [5]. The antioxidant defense system consisting of both antioxidant enzymes and non-enzymatic antioxidants plays an important role in scavenging reactive oxygen species (ROS) induced by severe drought stress. Previous studies have found that antioxidant capacity can be activated under drought stress to reduce ROS production and protect thylakoid membranes [6,7].

Brassinosteroids (BRs) are a kind of plant hormone that have an important effect on plant growth and development [8]. Previous studies have reported that BRs can alleviate the harmful effects of drought stress through increasing the activity of antioxidant enzymes and non-enzymatic antioxidant contents to eliminate ROS damage to membranes [9] or by elevating endogenous abscisic acid (ABA) to induce stomatal closure [10–12]. In addition, BRs can improve the energy metabolism balance between the chloroplast and mitochondria, boost the initial activity of Rubisco, and increase the utilization of light energy absorbed by plants, therefore increasing photosynthetic efficiency under drought stress [13,14]. However, the existing research results are mainly obtained from model plants, and little is known about non-model species.

Rhododendron, one of the most well-known ornamental plants, is widely distributed in Yunnan Province of southwestern China, which harbors more than 320 species [15]. Most of these rhododendrons are distributed in high-altitude areas, which commonly are less constrained by drought stress in the wild. However, the drought in winter (October–December) and spring (January–April) caused by a monsoon climate is an important factor restricting the growth and flowering performance of rhododendrons [16]. Among rhododendrons, *Rhododendron delavayi* Franch., is widely distributed throughout southwest China and grows at a wide altitudinal range, between 1200 and 3200 m [17]. Due to its large and scarlet flowers, *R. delavayi* has become a highly profitable ornamental flower in the market and is widely used in gardens, especially in China and some Southeastern Asian countries. In addition, *R. delavayi* showed a stronger ecophysiological performance in arid and high radiation environments. It was a representative species to study the drought tolerance of rhododendron [16,17]. Thus, understanding the tolerance mechanism of *R. delavayi* to drought stress is of great significance for cultivation and new variety breeding. In our previous study [18], we found that the BR signal transduction pathway may play an important role in improving drought tolerance of *R. delavayi* and the application of 24-epibrassionlide (EBR) improves its photosynthetic performance. However, the mechanisms of BRs improving the resistance of *R. delavayi* to drought stress remain obscure.

Proteomics approaches have been widely used to study the molecular mechanisms of plants adapting to drought stress. However, the proteomic analysis of alpine plants, such as rhododendron under drought stress, is very rare [19]. Thus, studying the proteomic changes under drought stress is of great significance for understanding the molecular response mechanisms of rhododendron to drought stress. Here, we compared the protein expression profiles under drought treatment with or without a pre-treat with BR to clarify the mechanisms of BRs regulating the response of *R. delavayi* to drought by employing a data-independent acquisition (DIA) proteomics approach. We hypothesized that: (1) application of EBR will alleviate the negative effect of drought stress; and (2) the key proteins and metabolic pathways related to improving photosynthesis, maintaining the balance between energy metabolism and carbon metabolism, and alleviating oxidative stress. Our findings offer new insights into the key responsive proteins and metabolic pathways involved in the adaptation of rhododendron to drought stress.

## 2. Materials and Methods

### 2.1. Plant Materials and Treatments

Three-year-old seedlings of *R. delavayi* which plant height is 30–40 cm, crown width is 15–25 cm, were used in our experiments, and 45 plants were grown in a volume of 9 L open plastic pots (one plant per pot) with a leaching hole. The pots filled with 10–40 mm "Pindstrup Sphagnum" peat moss in a greenhouse in Kunming, China (a1t. 1926 m, E 102°46′, N 25°07′). The growth conditions included an air temperature of 22–28 °C

(day) and 10–15 °C (night), 40–70% relative air humidity, and a maximum light intensity of approximately 1400 μmol m$^{-2}$s$^{-1}$. The plants were irrigated three times per week to maintain an adequate water supply. After 30 days, these 45 plants were randomly divided into three groups: control (CK), 24-epibrassionlide (EBR, Sigma, Saint Louis, MO, USA) (BR), and drought stress (D). Before commencement of the drought stress treatment, the plants of the BR group were sprayed with 1 L 1 mg L$^{-1}$ EBR [18] solution once a day for 5 days. The EBR solution was applied with a manual pump sprayer to cover the lower and upper leaf surface until dripping. At the same time, the plants of the CK and D groups were sprayed with the same amount of distilled water once a day for 5 days as the BR group did. After that, the plants of the BR and D groups were then subject to progressive drought stress, which was achieved by weighing the pots before irrigation and applying the required amount of water. The processing details were as follows: First, weigh the weight of each pot between 9:00–9:30 every morning and calculate the weight loss compared to the previous day. The weight loss was the amount of daily transpiration of each pot per day. After calculating the amount of daily transpiration of each pot, the plants of the CK group were supplemented with 100% of the amount of daily transpiration, and the plants of D and BR groups were supplemented with 30% of the amount of daily transpiration. Repeated the operation every day until the end of the experiment. Leaf samples of the CK, BR, and D groups used for proteomic analysis were collected at 8 and 18 d after drought treatment and named CK1, BR1, D1, and CK2, BR2, and D2, respectively. Each sample was pooled from five plants. Subsequently, the samples were immediately frozen in liquid nitrogen and stored at −80 °C until analysis. Three biological replicates were performed for each treatment.

## 2.2. Determination of Instantaneous Gas Exchange and Physiological Traits

Instantaneous gas exchange measurements were tested between 12:00 and 13:00 h every day using an open gas exchange system (Li-6400XT; Li-Cor, Inc., Lincoln, NE, USA) equipped with a light source (Li-6400-02B). All measurements were made on the fully expanded leaves at a saturating photosynthetic photon flux density (PPFD) of 1000 μmol m$^{-2}$s$^{-1}$, with a CO$_2$ concentration of 400 μmol mol$^{-1}$ in the leaf cuvette. The net photosynthetic rate (P$_n$), stomatal conductance (G$_s$), and transpiration rate (T$_r$) were recorded automatically by the Li-6400XT.

To evaluate the physiological changes of *R. delavayi* under different conditions, the content of malondialdehyde (MDA), soluble sugar and soluble protein content, and the activity of superoxide dismutase (SOD), peroxidase (POD), and catalase (CAT) were compared in the leaves of CK, BR, and D groups after the start of the experiment for 8 days. The leaves at the top of the branch were collected and each sample was measured with four replicates as previously described [18].

## 2.3. Protein Extraction and Digestion

The samples of CK1, BR1, D1, CK2, BR2, and D2 were subjected to DIA quantitative proteomic analysis by Gene Denovo Biotechnology Co. (Guangzhou, China). The protein extraction and digestion were according to the reported method with a minor modification [20]. Total proteins were extracted using the cold acetone method. In brief, the leaves for each sample were ground to powder in liquid nitrogen, then 0.5 g of the powder was collected and dissolved in 2 mL lysis buffer (8 M urea, 2% SDS, 1× Protease Inhibitor Cocktail (Roche Ltd., Basel, Switzerland)), followed by sonication on ice for 30 min and centrifugation at 13,000 rpm for 30 min at 4 °C The supernatant was collected and then transferred to a fresh tube. For each sample, the proteins were precipitated with ice-cold acetone at −20 °C overnight. The precipitation was cleaned with ice-cold acetone three times and re-dissolved in 8 M urea by sonication on ice.

The protein quality and concentration were tested by using sodium dodecyl sulfate-polyacrylamide gel electrophoresis (SDS-PAGE) and a BCA Protein Assay Kit, respectively. A 50 μg re-solubilized protein sample was suspended in 50 μL solution, reduced by adding

1 μL 1 M dithiotreitol at 55 °C for 1 h, alkylated by adding 5 μL 20 mM iodoacetamide in the dark at 37 °C for 1 h. The sample was then precipitated using 300 μL prechilled acetone at −20 °C overnight. The precipitate was washed twice with cold acetone and then re-suspended in 50 mM ammonium bicarbonate. Finally, the protein sample was digested with sequence-grade modified trypsin (Promega, Madison, WI, USA) at a substrate/enzyme ratio of 50:1 (*w/w*) at 37 °C for 16 h.

### 2.4. High pH Reverse Phase Separation

After protein digestion, the peptide mixture was re-dissolved in the buffer A (20 mM ammonium formate in water, pH 10.0, adjusted with ammonium hydroxide), and then fractionated by high pH separation using the Ultimate 3000 system (ThermoFisher Scientific, MA, USA) connected to a reverse-phase column (XBridge C18 column, 4.6 mm × 250 mm, 5 μm (Waters Corporation, Milford, MA, USA)). A linear gradient from 5% B to 45% B in 40 min (B: 20 mM ammonium formate in 80% ACN, pH 10.0, adjusted with ammonium hydroxide) was used for high pH separation. The column was re-equilibrated at the initial condition for 15 min. The column flow rate was 1 mL/min, and the column temperature was maintained at 30 °C. Ten fractions were collected and dried in a vacuum concentrator.

### 2.5. Nano-HPLC-MS/MS Analysis

For the nano-HPLC-MS/MS analysis, the peptide was re-dissolved in 30 μL solvent A (A: 0.1% formic acid in water), and analyzed by on-line nanospray LC-MS/MS on an Orbitrap Fusion Lumos (Thermo Fisher Scientific, Bremen, Germany) coupled to a Nano ACQUITY UPLC system (Waters Corporation, Milford, MA, USA). A 3 μL peptide sample was loaded onto the trap column (Acclaim PepMap C18, 75 μm × 25 cm; flow rate 300 nL/min) and was subsequently separated with a 120-min gradient, from 5% to 35% B (B: 0.1% formic acid in ACN). The column flow rate was maintained at 200 nL/min with column temperature of 40 °C, and the electrospray voltage was 2 kV. The mass spectrometer parameters were as follows: (1) MS: scan range ($m/z$) = 350–1200; resolution = 120,000; AGC target = 1,000,000; maximum injection time = 50 ms; (2) HCD-MS/MS: resolution = 30,000; AGC target = 1,000,000; maximum injection time = 35 ms; collision energy = 32; stepped CE = 5%.

### 2.6. Data Analysis

For analysis of differentially expressed proteins (DEPs), the raw DIA data were processed and analyzed by Spectronaut Pulsar 11.0 (Biognosys AG) with default parameters BGS Factory Settings. Proteins were annotated against the gene ontology (GO), Kyoto Encyclopedia of Genes and Genomes (KEGG), and Cluster of Orthologous Groups of Proteins (COG) databases to obtain their functions. Significant GO functions and pathways were examined within DEPs with a *p*-value of < 0.05, and a fold change of >1.5 or <0.67. Statistical analysis was performed with SPSS 16.0 (SPSS Inc., Chicago, IL, USA). One-way ANOVA and LSD multiple comparisons tests were performed to compare the effects of treatment (CK, BR, and D) on leaf physiological variables.

## 3. Results

### 3.1. Effects of BR and Drought on the Photosynthetic Performance of R. delavayi

The dynamic changes of $P_n$, $G_s$, and $T_r$ in *R. delavayi* under CK, BR, and D conditions were measured to examine the effects of BR and drought stress on photosynthesis. Overall, $P_n$ fluctuated greatly during the experiment (Figure 1A). At the beginning of drought stress, the $P_n$ value of the CK treatment was 11.7 μmol $CO_2$ m$^{-2}$s$^{-1}$, and gradually decreased to 7.8 μmol $CO_2$ m$^{-2}$s$^{-1}$ after 18 days. The BR-treated plants showed a similar change pattern and photosynthetic rate as the CK plants, while the $P_n$ values of drought-stressed plants decreased down to 5.0 μmol $CO_2$ m$^{-2}$s$^{-1}$ on the 18th day after treatment.

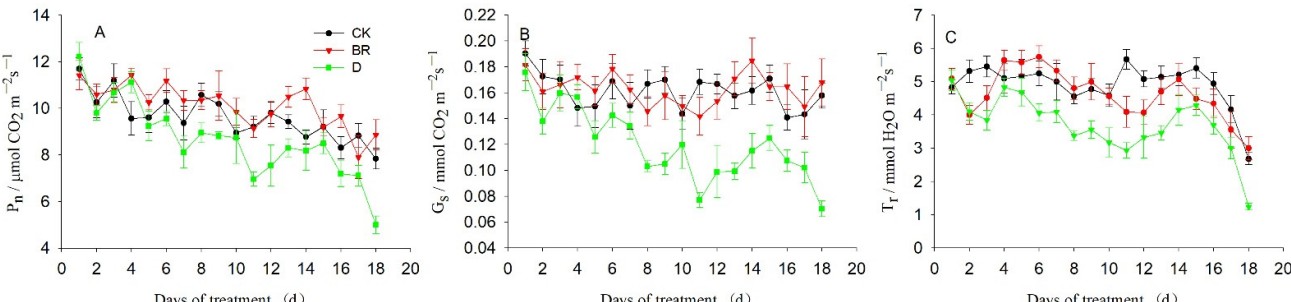

**Figure 1.** Variations of (**A**) net photosynthetic rate ($P_n$), (**B**) stomatal conductance ($G_s$) and (**C**) transpiration rate ($T_r$) under control (CK), drought stress with 24-epibrassionlide (BR) pre-treatment, and without (D) BR pre-treatment. Values represent the means $\pm$ SE ($n$ =10).

Although $G_s$ and $T_r$ showed a similar variation to $P_n$, the changes of $G_s$ and $T_r$ were more stable than $P_n$ during treatment. In the CK and BR treatments, $G_s$ were maintained above 0.14 mol $CO_2$ m$^{-2}$s$^{-1}$. In the D treatment, $G_s$ were close to 0.14 mol $CO_2$ m$^{-2}$s$^{-1}$ in the first 7 days after treatment but significantly declined from the 8th day after treatment. The values of $G_s$ and $T_r$ in the drought-stressed plants were less than 50% of the CK plants on the 18th day after treatment (Figure 1B,C).

*3.2. Effects of BR and Drought on Physiological and Biochemical Substances of R. delavayi*

The MDA content, soluble sugar content, soluble protein content, and SOD, POD, and CAT activity increased in the D and BR treatments compared with the CK treatment, but the increment in the BR treatment was less than that of the D treatment. The MDA content, soluble sugar content, soluble protein content, and SOD activity under the D treatment were significantly higher than that of the CK and BR treatments, while the changes in POD and CAT activity were not significant among treatments (Figure 2).

*3.3. Quantitative Proteomic Analysis*

A total of 25,904 peptides, 19,474 unique peptides, and 3453 proteins (Supplementary Table S1) were identified by DIA analysis across two time points of the CK, BR, and D groups (Figure 3A). Most of the peptides ranged from 8 to 20 amino acids (Figure 3B), with 83.3% (16,070) of the peptides weighing 1000–2500 Da (Figure 3C). Over 81.6% of the total proteins (3261) matched with at least two peptides (Figure 3D). These proteins were divided into 25 categories. Among them, the general function prediction contained the largest number of proteins (14.4%), followed by posttranslational modification, protein turnover, chaperones (14.3%), translation, ribosomal structure and biogenesis (9.2%), and energy production and conversion (7.9%) (Figure 4).

*3.4. Identification of Differentially Expressed Proteins (DEPs)*

Based on the fold change >1.5 (up) or <0.67 (down) with $p$-value < 0.05, 683 DEPs were identified (Supplementary Table S2). Compared with the CK1, the expression levels of 125 (75 up/50 down) and 32 (9 up/23 down) proteins showed significant changes in the BR1 and D1 groups, respectively (Figure 5A). Of the 144 proteins with significant changes, 13 (9.03%, 7 up/6 down) were commonly found in the BR1 and D1 groups, while 112 (77.78%, 68 up/44 down) and 19 (13.19%, 2 up/17down) proteins were exclusively identified in the BR1 and D1 groups, respectively (Figure 5B). Compared with the CK2, the expression levels of 445 (238 up/207 down) and 396 (247 up/149 down) proteins changed significantly in the BR2 and D2 groups, respectively. Of the 609 proteins with significant changes, 232 (38.10%, 150 up/82 down) were commonly found in the BR2 and D2 groups, while 213 (35.26%, 88 up/125 down) and 164 (26.93%, 97 up/67 down) proteins were exclusively found in the BR2 and D2 groups, respectively (Figure 5C). Overall, more

DEPs were found in the BR treatment than in the D treatment, and most of the DEPs were exclusively found in the BR and D treatments.

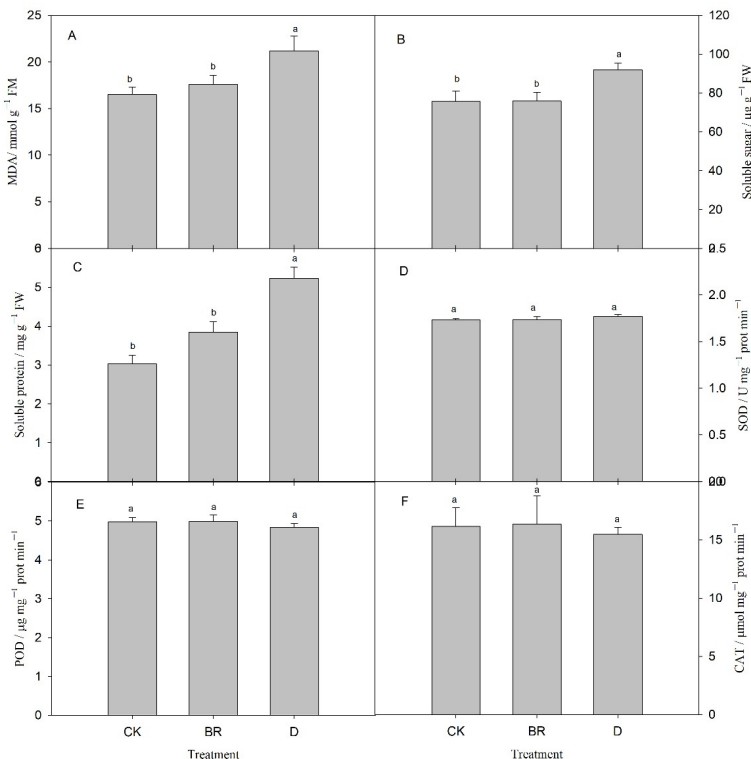

**Figure 2.** Physiological characteristics of *Rhododendron delavayi* under control (CK), 24-epibrassionlide (BR), and drought (D) conditions. Data are the means $\pm$ SE ($n = 4$). Different lowercase letters indicate a significant difference at the 5% level; (**A**) leaf malondialdehyde (MDA) content; (**B**) soluble sugar content; (**C**) soluble protein content; (**D**) superoxide dismutase (SOD) activity; (**E**) peroxidase (POD) activity; (**F**) catalase (CAT) activity.

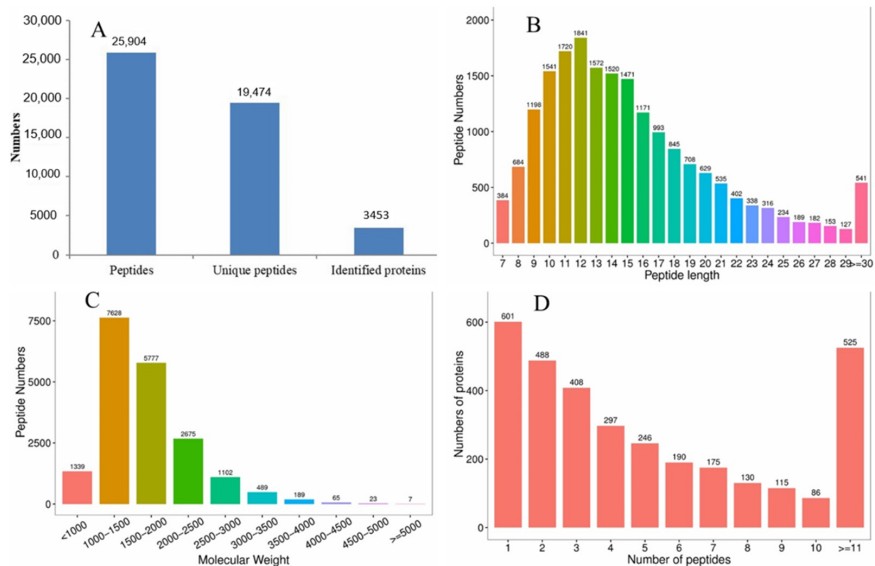

**Figure 3.** Basic Data-Independent Acquisition (DIA) proteomics analysis output details. (**A**) Peptides and protein species identified by a DIA database search; (**B**) length distribution of peptides; (**C**) molecular weight of peptides; (**D**) numbers of peptides that were matched to proteins.

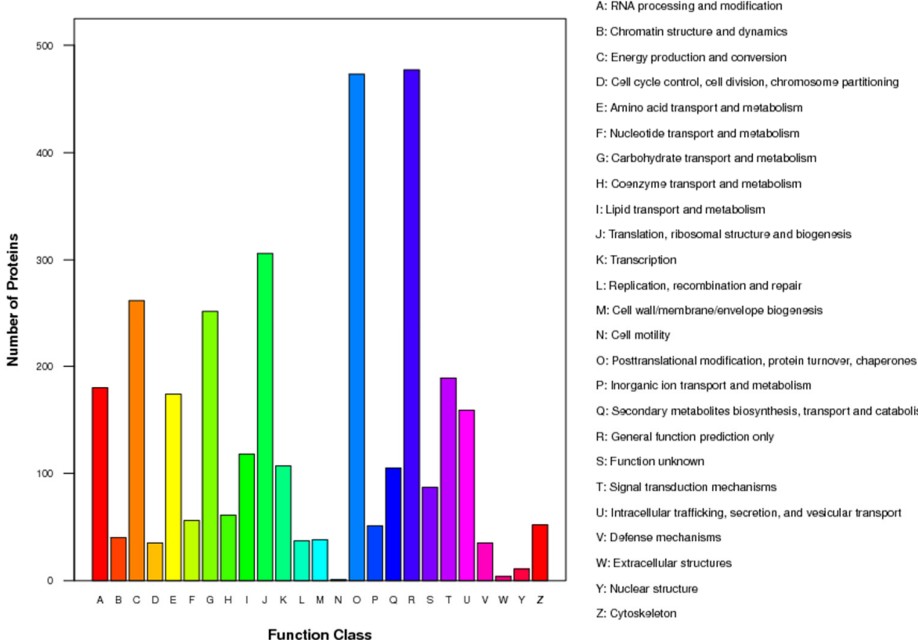

**Figure 4.** Cluster of Orthologous Groups of Proteins (COG) annotation analysis of all proteins.

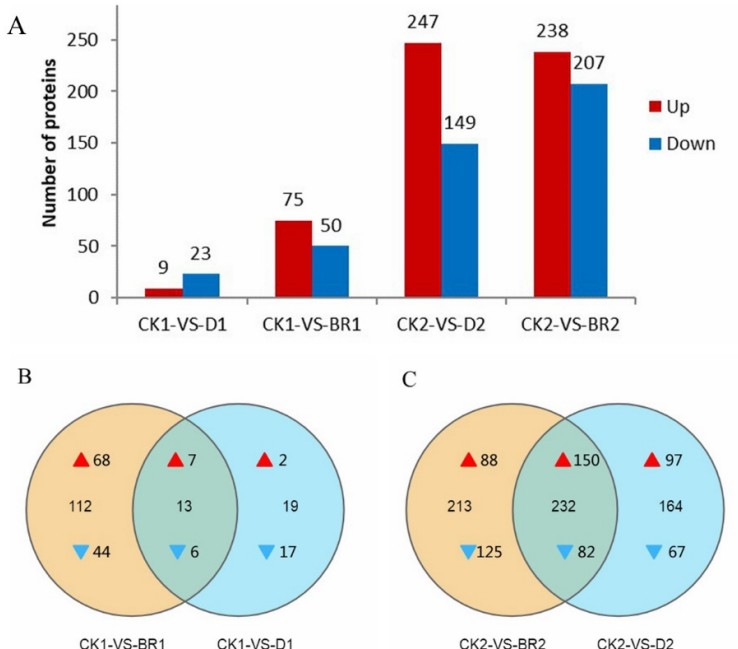

**Figure 5.** Number of differentially expressed proteins (DEPs) identified between groups (**A**), and Venn diagrams showing the number of DEPs between 24-epibrassionlide (BR) and drought (D) conditions at the early phase (**B**) and late phase (**C**), respectively. The numbers attached behind the sample identifiers represent the time point at which the samples were collected: 1 and 2 represent 8 and 18 days of treatment, respectively.

### 3.5. Functional Classification of DEPs

All DEPs were classified into three functional groups: biological process, cellular component, and molecular. Most of the DEPs with a biological process were enriched in the BR and D treatments, followed by the cellular component and molecular function. On the one hand, some GO terms were shared between the BR and D treatments (Figures 6 and 7A). Within the biological process group, most of the annotated categories were enriched in metabolic process, cellular process, and single-organism process. In terms of the cellular

component, proteins were highly enriched in the cell, cell part, and organelle, while most of the annotated molecular functions were related to catalytic activity and RNA binding. On the other hand, more unique function categories were identified in the BR treatment than that of the D treatment, including immune system process, positive regulation of biological process, detoxification, extracellular region, electron carrier activity, molecular transducer activity, and signal transducer activity (Figure 6A). This suggested that the proteins in these categories took part in BR-induction responses. Only the category of biological adhesion was especially identified in the drought-induced proteins (Figure 7A).

The ribosome, ether lipid metabolism, photosynthesis, and oxidative phosphorylation pathways were the most significantly enriched in the BR treatment. Contrastingly, the flavonoid biosynthesis, ubiquitin-mediated proteolysis, biosynthesis of secondary metabolites, and alpha-linolenic acid metabolism were the most significantly enriched in the D treatment. The shared proteins in the BR and D treatments included starch and sucrose metabolism, biosynthesis of secondary metabolites, aminoacyl-tRNA biosynthesis, alpha-linolenic acid metabolism, pentose phosphate pathway, tyrosine metabolism, linoleic acid metabolism, carbon metabolism, glycolysis/gluconeogenesis, and MAPK signaling pathway (Table 1).

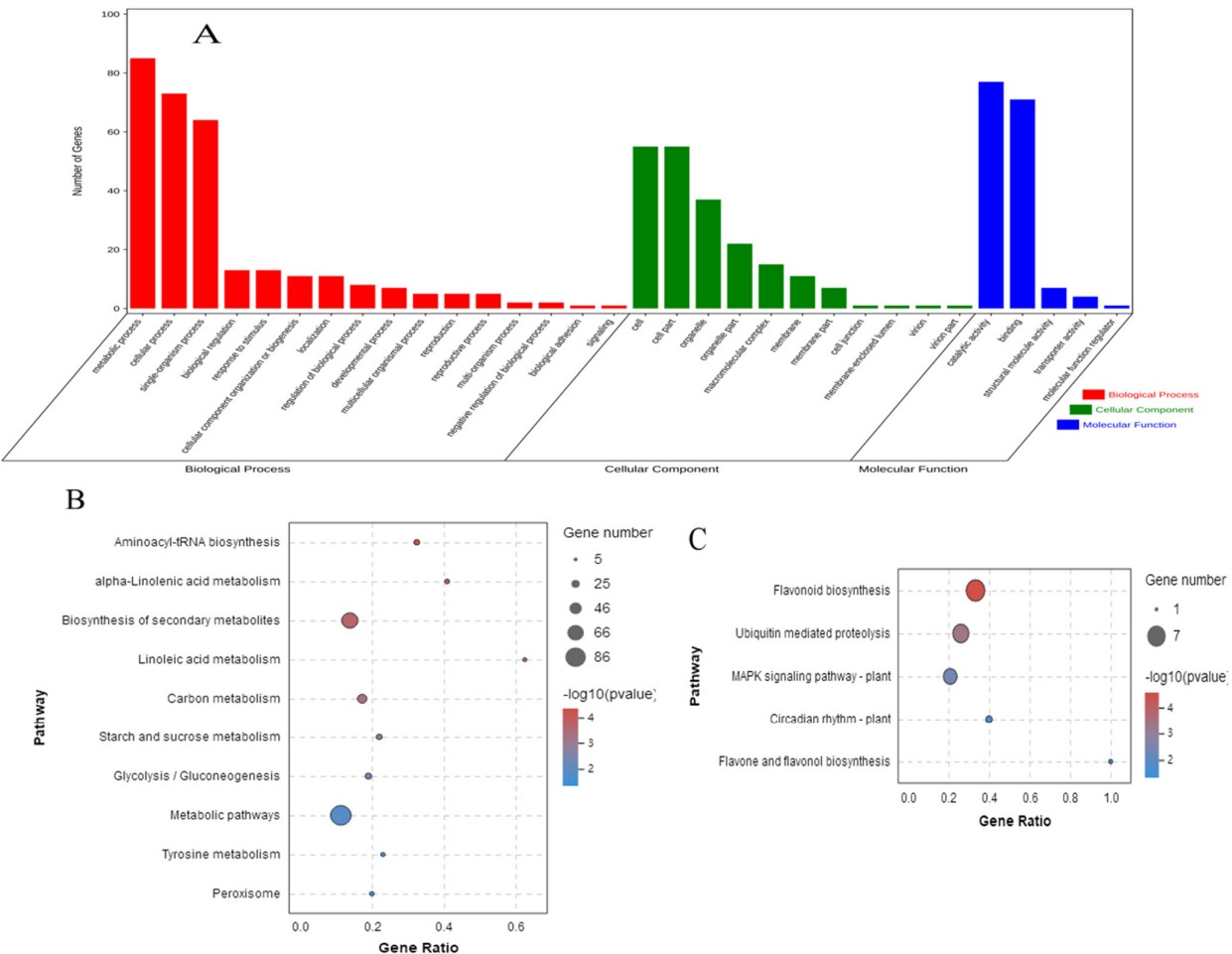

**Figure 6.** (**A**) GO-based enrichment analysis of differentially expressed proteins (DEPs) in leaves of *Rhododendron delavayi* under drought stress; (**B**) Significantly enriched (*p* < 0.05) pathway of up-accumulated DEPs under drought stress; (**C**) Significantly enriched (*p* < 0.05) pathway of down-accumulated DEPs under drought stress.

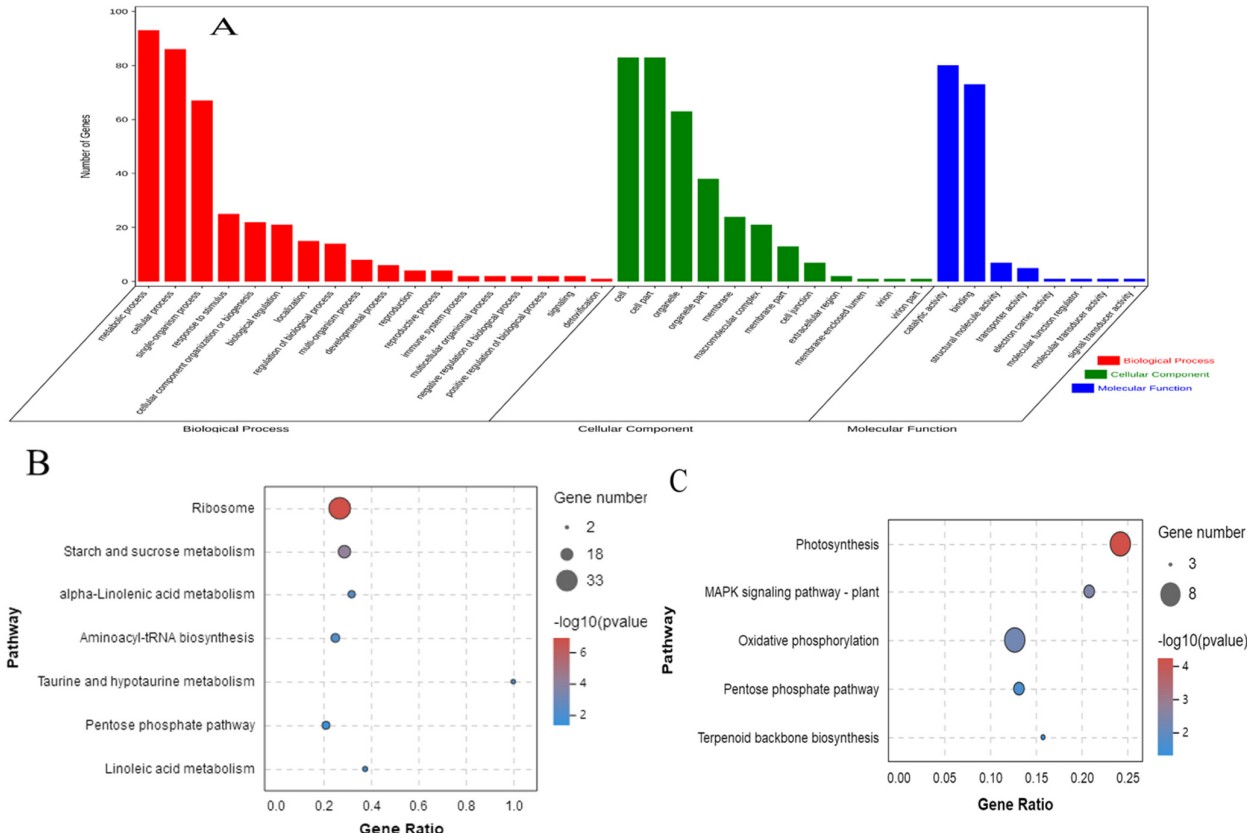

**Figure 7.** (**A**) GO-based enrichment analysis of differentially expressed proteins (DEPs) in leaves of *Rhododendron delavayi* under BR treatment; (**B**) Significantly enriched ($p < 0.05$) pathway of up-accumulated DEPs under BR treatment; (**C**) Significantly enriched ($p < 0.05$) pathway of down-accumulated DEPs under BR treatment.

**Table 1.** Significantly enriched KEGG pathways of differentially expressed proteins (DEPs) specific to 24-epibrassionlide (BR) treatment, drought (D) treatment, and shared between the BR and D treatments.

| Pathway | DEPs with Pathway Annotation | All Proteins with Pathway Annotation | *p*-Value | Q-Value | Pathway ID |
|---|---|---|---|---|---|
| **Specific to BR treatment** | | | | | |
| Ribosome | 28 | 123 | 0 | 0 | ko03010 |
| Ether lipid metabolism | 2 | 2 | 0.006 | 0.172 | ko00565 |
| Photosynthesis | 7 | 33 | 0.009 | 0.187 | ko00195 |
| Oxidative phosphorylation | 9 | 63 | 0.041 | 0.627 | ko00190 |
| **Specific to D treatment** | | | | | |
| Flavonoid biosynthesis | 6 | 21 | 0.001 | 0.088 | ko00941 |
| Ubiquitin mediated proteolysis | 5 | 23 | 0.012 | 0.268 | ko04120 |
| Biosynthesis of secondary metabolites | 40 | 468 | 0.012 | 0.268 | ko01110 |
| Alpha-linolenic acid metabolism | 4 | 22 | 0.045 | 0.741 | ko00592 |
| **Shared between BR and D treatments** | | | | | |
| Starch and sucrose metabolism | 12 | 59 | 0.003 | 0.110 | ko00500 |
| Biosynthesis of secondary metabolites | 53 | 468 | 0.004 | 0.110 | ko01110 |
| Aminoacyl-tRNA biosynthesis | 9 | 40 | 0.004 | 0.110 | ko00970 |
| Alpha-linolenic acid metabolism | 6 | 22 | 0.007 | 0.140 | ko00592 |
| Pentose phosphate pathway | 8 | 38 | 0.011 | 0.165 | ko00030 |
| Tyrosine metabolism | 6 | 26 | 0.017 | 0.218 | ko00350 |
| Linoleic acid metabolism | 3 | 8 | 0.023 | 0.255 | ko00591 |
| Carbon metabolism | 22 | 179 | 0.033 | 0.321 | ko01200 |
| Glycolysis/Gluconeogenesis | 12 | 84 | 0.041 | 0.334 | ko00010 |
| MAPK signaling pathway-plant | 5 | 24 | 0.043 | 0.334 | ko04016 |

The KEGG analysis showed that the highly abundant protein species were mainly involved in aminoacyl-tRNA biosynthesis, alpha-linolenic acid metabolism, biosynthesis of secondary metabolites, carbon metabolism, starch and sucrose metabolism, glycolysis/gluconeogenesis, tyrosine metabolism, and peroxisome in the D treatment, (Figure 6B), while those with low abundance protein species were mainly involved in flavonoid biosynthesis, ubiquitin-mediated proteolysis, MAPK signaling pathway, and circadian rhythm (Figure 6C). Most of the highly abundant protein species in the BR treatment were involved in similar metabolism pathways as those in the D treatment. However, the ribosome category was only highly enriched in the BR treatment (Figure 7B). The low abundance protein species in the BR treatment were involved in photosynthesis, MAPK signaling pathway, oxidative phosphorylation, pentose phosphate pathway, and terpenoid backbone biosynthesis (Figure 7C).

## 4. Discussion

### 4.1. Response of Photosynthetic and Physiological Performances to Drought

As one of the most critical metabolic processes, photosynthesis is primarily affected by drought [21]. Previous studies have found that exogenous BRs can ameliorate the stress-induced inhibition of photosynthesis [22,23]. In the present study, exogenous BR application improved the photosynthetic performance of *R. delavayi* under drought conditions. It was likely that EBR improved the efficiency of photosynthetic carbon fixation by alleviating stomatal closure (Figure 1). High stomatal conductance increases $CO_2$ supply for carbon assimilation and fixation [24]. In addition, our previous study showed that BR-induced improvement in $CO_2$ assimilation of *R. delavayi* under drought is related to the increase in light utilization efficiency and photochemical efficiency [18]. Thus, the improvement of photosynthetic performance may be one of the most important characteristics of BR improving the drought resistance of *R. delavayi*.

Under drought stress, the stimulated ROS production induces lipid peroxidation, therefore resulting in electrolyte leakage and membrane damage [25]. The MDA is the final product of the decomposition of membrane peroxidation. The MDA is regarded as a marker for assessing lipid peroxidation or damage to cytoplasmic and organelles membranes [26]. Our study showed that MDA concentration increased significantly under drought, whereas the level of MDA was lower in BR treatment (Figure 2). This indicates that pretreatment with BR better protects *R. delavayi* against oxidative damage under drought. Plants can eliminate the toxicity of ROS by employing endogenous antioxidant enzymes and non-enzymatic antioxidants [6,7]. A previous study has shown that BR treatment leads to an increase in antioxidant enzymes, such as SOD, APX, and CAT, and can eliminate the toxicity of ROS under drought stress [27]. From our findings, the activities of SOD, POD, and CAT increased in the BR treatment. A previous study showed that the oxidative damage estimated by MDA accumulation was lower in response to drought which is directly linked with a higher expression level of antioxidant enzymes in wheat [28]. In the present study, the increase in SOD, POD, and CAT could be related to an active and efficient antioxidant response that might be involved in maintaining a lower MDA concentration under drought and therefore helping *R. delavayi* to cope with drought stress. In addition, under drought conditions, the application of BR can effectively improve plant water relations by maintaining leaf water potential, alleviating the effect of drought stress on stomatal close, and improving photosynthesis under drought [18]. Thus, pre-treatment with BR may increase the tolerance of *R. delavayi* to drought stress by ameliorating photosynthetic depression caused by stomatal closure, improving water relations, and decreasing oxidative stress.

### 4.2. Significant Changes of Ribosomal Proteins in the BR Treatment

There was a pronounced change in the drought-responsive proteome in the BR treatment. Among these DEPs, the ribosome category was only highly enriched in the BR treatment. Ribosomal proteins (RP) are well known for their universal roles in forming

and stabilizing the ribosomal complex and mediating protein synthesis [29]. Commonly, RP genes have roles not only in ribosome biogenesis and plant growth and development but also in abiotic and biotic stress responsiveness and tolerance [30]. For example, in a previous study, RPL10 was significantly up-regulated in response to UV-B radiation [31] or drought stress. In another study, overexpression of RPL23A significantly increased the fresh weight, root length, proline, and chlorophyll contents of rice under drought and salt stresses [29]. In the present study, a total of 45 DEPs involved in the ribosome pathway were identified, and 75.56% of these RPs were significantly upregulated under drought stress, including the RP large subunit and the RP small subunit (Supplementary Table S3). The significant upregulation of ribosomal components under drought stress indicated that translation of some, perhaps highly stable and/or RNA-binding proteins, were an important component of the BR regulating drought-response of *R. delavayi*.

### 4.3. Proteins in Relation to Energy Metabolism

The DEPs involved in energy metabolism are summarized in Supplementary Table S4. The pathways related to photosynthesis and oxidative phosphorylation were the most enriched metabolic pathways in the BR treatment. Among 11 DEPs involved in photosynthesis, 9 proteins (Lhca3, PsaN, PetC, 3PetF, PsbO, PsbP, ATPF1D) involved in light-harvesting chlorophyll–protein complex (LHC), photosystem I (PSI), cytochrome complex, photosynthetic electron transport, and ATP synthase were downregulated. The LHC plays an important role in light harvesting in PSI and photosystem II (PSII). A low level of LHC protein affects the capture and transmission of light energy in the chloroplast [32]. In the present study, the suppression of the proteins involved in PSI, cytochrome complex, and photosynthetic electron transport indicated that the binding stability of PSI and the rate of electron transfer to the photosynthetic center in *R. delavayi* was inhibited by drought. The decrease in photosynthetic capacity under drought stress (Figure 1) verified this result.

Among the nine DEPs involved in carbon fixation, seven proteins were increased in abundance, including glyceraldehyde 3-phosphate dehydrogenase (NADP$^+$) (GAPD), fructose-1,6-bisphosphatase (FBP), transketolase, phosphoenolpyruvate carboxylase (PPC), orthophosphate dikinase (PPDK), and aspartate aminotransferase (GOT1). Glyceraldehyde 3-phosphate dehydrogenase (GAPD) is a key enzyme in carbon fixation and catalyzes 1,3-biophosphoglycerafe to 3-phosphate glyceraldehyde. Fructose-1,6-bisphosphatase (FBP) is a key enzyme that transfers fructose-1,6-bisphosphate to fructose-6-bisphosphate [33]. The high abundance of these protein species showed that *R. delavayi* maintained strong $CO_2$ fixation capacity under drought stress. Thus, the photosynthesis limitation of *R. delavayi* mainly occurred in the light reaction stage rather than the dark reaction stage under drought stress.

A total of 14 DEPs were involved in the oxidative phosphorylation process. These DEPs were NADH dehydrogenases, NADH ubiquinone oxidoreductases, and other enzymes involved in ATP synthesis. NADH ubiquinone oxidoreductases can generate superoxide and $H_2O_2$ through multiple pathways, which are an important source of ROS [34]. In the present study, the upregulated expression of NADH ubiquinone oxidoreductases suggested that the production of ROS was accelerated under drought stress. However, mitochondria can prevent ROS generation via employing the alternative oxidase (AOX) pathway, in which complexes III and IV are bypassed and electrons are directly transferred to oxygen, resulting in thermal energy being generated instead of ATP [35]. We observed that some proteins associated with ATP biosynthesis including ATP synthase subunit delta and subunit d were downregulated, while other proteins including ATP synthase subunit alpha, subunit O and NADH dehydrogenase were upregulated during drought stress. The downregulated expression of ATP biosynthesis proteins suggested that the energy produced was in the form of thermal energy to avoid ROS accumulation. Meanwhile, pretreatment with BR enabled *R. delavayi* to maintain a higher energy metabolism for normal growth. Thus, we inferred that the uncoupling of oxidative phosphorylation was one of the main strategies to reduce the accumulation of ROS under drought stress. Maintaining

the energy balance between thermal dissipation and ATP biosynthesis was an important mechanism for improving the drought resistance of *R. delavayi* through the BR treatment.

### 4.4. Carbohydrate Metabolism Is Important for R. delavayi to Cope with Drought Stress

Carbohydrate metabolism, including starch and sucrose metabolism, pentose phosphate, glycolysis/gluconeogenesis, and citrate cycle (TCA cycle), was significantly enriched in the D and BR treatments (Table 1). This indicated that carbohydrate metabolism was an important metabolic pathway for *R. delavayi* to cope with drought stress. In the glycolysis/gluconeogenesis pathway, the expression of proteins including hexokinase (HK), 6-phosphofructokinase (PFK), fructose-6-phosphate 1-phosphotransferase (PFP), pyruvate kinase (PK), alcohol dehydrogenase (ADH), pyruvate decarboxylase (PDC), and aldehyde dehydrogenase (NAD$^+$) were increased in the D and BR treatments (Supplementary Table S5), resulting in the abundant production of ATP and NADPH. Among these proteins, HK is a major rate-limiting enzyme in the glycolysis pathway. In the present study, the up-regulated expression of HK protein both in the D and BR treatments indicated that the HK protein produces more glucose-6-phosphate to facilitate the follow-up process of glycolysis under drought stress.

The interconversion between fructose-6-phosphate and fructose-1,6-bisphosphate is catalyzed by PFP and PFK, but the reaction catalyzed by PFP is reversible, and the reaction catalyzed by PFK is irreversible [36]. In our study, PFP and PFK were significantly up-regulated in the D and BR treatments. The increase in PFP and PFK expression enhanced the rate of glycolysis and promoted the production of energy through the glycolysis pathway. In addition, the expression of FBP also increased. This protein catalyzes fructose-1,6-bisphosphate convert into fructose-6-phosphate. Thus, the increase in FBP expression promotes the generation of glucose through the gluconeogenesis pathway.

Pyruvate kinase (PK) is one of the main rate-limiting enzymes in the conversion of phosphoenolpyruvate and ADP into ATP and pyruvate acid [36]. In our study, the up-regulated expression of the PK family proteins in the D and BR treatments (Supplementary Table S5) enabled *R. delavayi* to produce more energy via glycolysis under drought stress. Most of the proteins in starch and sucrose metabolism (90.48%), pentose phosphate (61.54%), and the tricarboxylic acid (TCA) cycle (100%) pathway were upregulated. The high expression of these enzymes increased the production of energy by increasing ATP biosynthesis, improving the connection between glycolysis and starch and sucrose metabolism, pentose phosphate, and the TCA cycle. Thus, carbohydrate metabolism is critical for meeting the energy demand for the survival of *R. delavayi* under drought stress and plays a key role in BR regulating the adaptation of *R. delavayi* to drought.

### 4.5. Proteins Involved in Cell Detoxification

Drought stress can stimulate ROS production. Excess ROS induces the peroxidation of membrane lipids, leading to the accumulation of aldehydes and damage to the cell membrane [25]. Aldehyde dehydrogenase can reduce the accumulation of aldehydes in plants by catalyzing the oxidative dehydrogenation of aldehydes to generate corresponding carboxylic acids by combining NAD$^+$ or NADP$^+$, to achieve their detoxification purposes. In addition, aldehyde dehydrogenase and alcohol dehydrogenase are involved in the catalytic decomposition of ethanol, which is eventually oxidized into $CO_2$ and $H_2O$, providing an intermediate product for plant metabolism [37]. In this study, aldehyde dehydrogenase and alcohol dehydrogenase were upregulated (Supplementary Table S6), indicating that these proteins may play an important role in eliminating the toxicity of aldehydes under drought stress.

Isocitrate dehydrogenase (IDH), 6-phosphogluconate dehydrogenase (PGD), and glutathione reductase (GSR) in the glutathione metabolism pathway were upregulated both in the D and BR treatments, and the expressions of IDH, PGD, and GSR were higher in the BR treatment than in the D treatment. The upregulation of GSR increases the content of glutathione (GSH), which has the function of eliminating ROS and lipid hydroperoxides [38].

The results indicated that the GSH antioxidant system for eliminating ROS production is involved in the response of *R. delavayi* to drought.

*4.6. Proteins Related to Lipid Metabolism*

Lipids not only provide energy for plant growth, but also participate in physiological reactions in plants, particularly in stabilizing cell membranes, and responding to drought, chilling, and salt stress. The total lipid content decreases under drought stress due to the inhibition of lipid synthesis and lipid decomposition and peroxidation [39]. In the present study, the proteins involved in linoleic acid metabolism and α-linolenic acid metabolism were significantly up-regulated in the D and BR treatments, and the increase in abundance in the BR treatment was greater than in the D treatment (Supplementary Table S7). Zafari et al. [40] found that foliar EBR application enhanced lipid metabolism of safflower under drought stress. In response to biotic and abiotic stresses, plants can remodel membrane fluidity by releasing α-linolenic acid. A previous study has found that free α-linolenic acid can alleviate the inhibition of drought on the growth of roots and seedlings, and the alleviation effect is enhanced with the increase of drought intensity [41]. Thus, we hypothesize that BRs-induced improvements in plant drought resistance may relate to the accumulation of lipids.

*4.7. Lignin Synthesis in the Response of R. delavayi to Drought Stress*

Some secondary metabolite biosynthesis-related proteins were up-regulated in the D and BR treatments, indicating that the synthesis of secondary metabolites increased under drought stress. Among the eight DEPs related to phenylpropanoid biosynthesis, 75% of these proteins were significantly upregulated, and the expression level in the BR treatment was higher than in the D treatment (Supplementary Table S8). High expression of cinnamyl-alcohol dehydrogenase (CAD) promotes more caffeyl-aldehyde and coniferyl-aldehyde converted into caffeyl-alcohol and coniferyl- alcohol. Caffeyl-alcohol and coniferyl- alcohol are precursors of lignin synthesis. Lignin is the main component of the cell wall, participating in enhancing plant resistance to drought [42]. Increased lignin content during drought stress might maintain the normal osmotic pressure of the cells, thereby enhancing the drought resistance of *R. delavayi*.

*4.8. Roles of Flavonoid Biosynthesis Proteins in Drought*

Flavonoids can scavenge ROS in response to drought stress and protect cells from oxidative damage [43]. In this study, seven proteins involved in flavonoid biosynthesis were identified (Supplementary Table S8). The expression levels of chalcone synthase (CHS), naringenin 3-dioxygenase (F3H), and anthocyanidin synthase (ANS) were significantly downregulated after exposure to drought stress. Chalcone synthase (CHS), F3H, and ANS are the key enzymes in flavonoid biosynthesis. Short-term drought stress can increase the expression abundance of these enzymes [44], while long-term drought suppresses their transcript abundance and activity [45,46]. The genes encoding CHS, F3H, and ANS were significantly up-regulated after *R. delavayi* was exposed to drought stress [47]. Combined with the present study, we found that the mRNA and protein levels exhibited inconsistent expression changes, suggesting that post-transcriptional regulation may affect the expression regulation of flavonoid biosynthesis genes in *R. delavayi*. Previous studies have also demonstrated that the gene expression of flavonoid biosynthesis is governed by the transcriptional and post-transcriptional regulation in response to environmental stimuli [48,49]. In the future, a comprehensive analysis of transcriptome and proteomics is needed to reveal the role of post-transcriptional regulation in the response of *R. delavayi* to drought stress.

**5. Conclusions**

In this study, we used a DIA approach to identify the protein expression profiles in *R. delavayi* under drought stress with or without a BR pre-treatment. The plants with the BR

pre-treatment exhibited stronger drought tolerance than those without the BR pre-treatment. A series of physiological mechanisms were involved in the BR-regulated response of *R. delavayi* to drought stress. However, improving photosynthesis, maintaining the integrity and stability of ribosomal complex to mediate protein synthesis, maintaining the balance between energy metabolism and carbon metabolism, and alleviating oxidative stress played more key roles in enhancing drought tolerance (Figure 8). Our study presents a comprehensive understanding of the proteins and metabolic pathways related to the response of *R. delavayi* to drought and will contribute to the breeding of drought-tolerant rhododendrons.

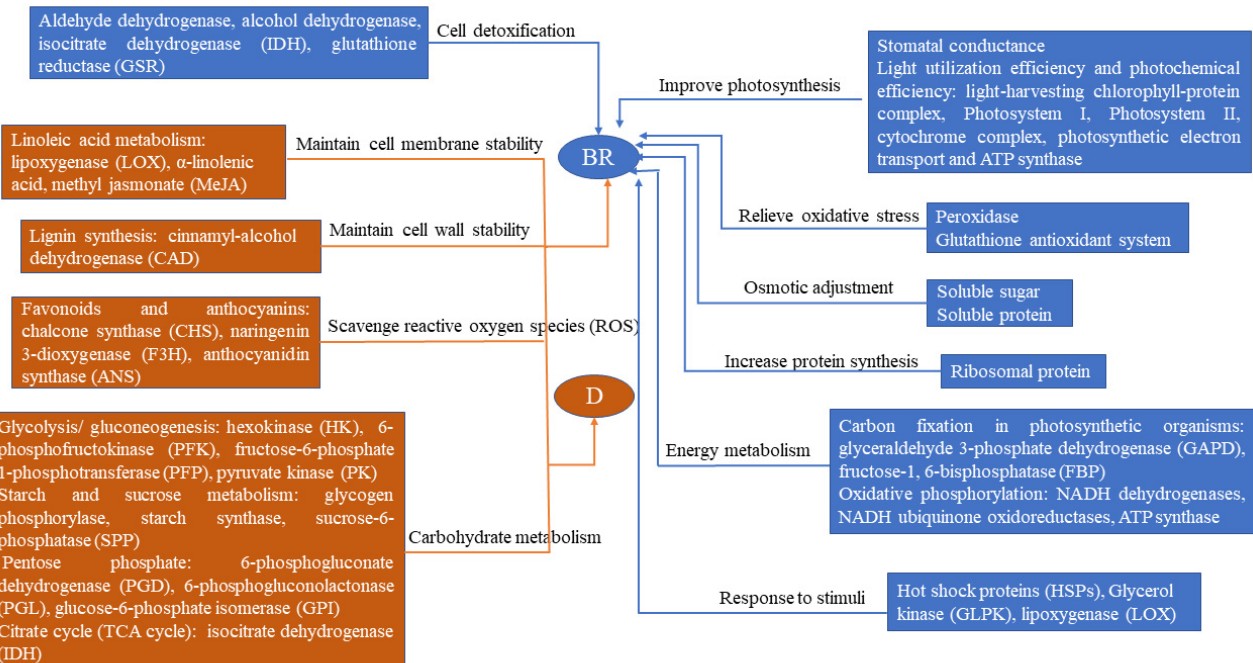

**Figure 8.** Model of BR regulating the response of *Rhododendron delavayi* to drought.

**Supplementary Materials:** The following are available online at https://www.mdpi.com/article/10.3390/horticulturae7110501/s1, Table S1: The total proteins that were identified by DIA, Table S2: All information of differentially expressed proteins (DEPs) identified between groups, Table S3: The differentially expressed proteins (DEPs) involved in ribosome with significant change in abundance at 8 and 18 days of drought (D) and BR pre-treatment (BR), Table S4: The differentially expressed proteins (DEPs) involved in energy metabolism with significant change in abundance at 8 and 18 days of drought (D) and BR treatment (BR), Table S5: The differentially expressed proteins (DEPs) involved in carbohydrate metabolism with significant change in abundance at 8 and 18 days of drought (D) and BR treatment (BR), Table S6: The differentially expressed proteins (DEPs) involved in cell detoxification with significant change in abundance at 8 and 18 days of drought (D) and BR treatment (BR), Table S7: The differentially expressed proteins (DEPs) involved in lipid metabolism with significant change in abundance at 8 and 18 days of drought (D) and BR treatment (BR), Table S8: The differentially expressed proteins (DEPs) involved in biosynthesis of secondary metabolites and other secondary metabolites with significant change in abundance at 8 and 18 days of drought (D) and BR treatment (BR).

**Author Contributions:** Conceptualization, Y.-F.C.; formal analysis, L.Z.; funding acquisition, Y.-F.C., L.Z. and L.-C.P.; investigation, S.-F.L.; methodology, Y.-F.C., L.Z., L.-C.P., J.S. and W.-J.X.; supervision, J.-H.W.; validation, J.S.; writing—review and editing, J.-H.W. All authors have read and agreed to the published version of the manuscript.

**Funding:** This research was financially supported by the National Natural Science Foundation of China (31760229, 31760230, 31760231), Construction of International Flower Technology Innovation Center and Industrialization of Achievements (2019ZG006). Science and Technology Talent and Platform Program of Yunnan Province (202105AC160017).

**Institutional Review Board Statement:** Not applicable.

**Informed Consent Statement:** Not applicable.

**Data Availability Statement:** Data is available by contact with the first author.

**Conflicts of Interest:** The authors declare no conflict of interest.

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
