# Peer review of "Key Proteins and Metabolic Pathways Involved in 24-Epibrasionlide Improving Drought Tolerance of Rhododendron delavayi Franch"

_horticulturae, doi:10.3390/horticulturae7110501_

Round 1

Reviewer 1 Report

In this paper, Cai et al studied the Drought-responsive proteins and metabolic pathways involved in 24-epibrassionlide regulating drought tolerance of Rhododendron delavayi. To accomplish this, the authors have checked the various physiological and biochemical parameters of Rhododendron plants grown under drought-stress and control conditions. The authors demonstrated that BR activating the numerous physiological and biochemical changes to alleviate the effects of drought-stress. Finally, they proposed the use of BR in drought-stress management in plants. However, I have major concerns about the methodology.

Author Response

Response to Reviewer 1 Comment

Introduction

(1) Line 37-39: The authors are suggested to cite more recent articles instead of old references.

Response: We have cited the more recent articles instead of old references (Revision: Line 37-38, Reference 1,2,3).

Material and methods:

(1) Line 85: This section needs to improve. Better to be more descriptive。

Response: We have improved the section of ‘Plant material and treatments’ (Revision: Line 90-110).

(2) Line 86: Please explain more about plants such as height and ...

Response: We added some information about plant materials (Revision: Line 90).

(3) Line 92: Provide the detailed characteristics of BR used in this study.

Response: We added information about the BR that used in this study (Revision: Line 97).

(4) Line 92: Please explain the methodology of stress application.

Response: We added more details about the methodology of stress application (Revision: Line 103-110).

(5) Line105: The authors didn’t provide the information that which leaf/leaves (position on plants), they have used for biochemical analysis.

Response: The leaves at the top of the branch were collected and used for biochemical analysis. We add the information about its (Revision: Line 122-126).

Discussion

(1) Line 279: What was the correlation between BE and DROUGHT in influencing the plant health?

Response: We revised the line 304-308, and line 311-319. Also, we added some references to support our opinion (Revision: Reference 6, 7, 26, 28).

(2) Line 451: Authors provide explanation if any known pathway or mechanism how the BR provide tolerance to DROUGHT stress by binding to any solutes or oxidative enzymes?

Response: From our findings, solutes and oxidative enzymes changed significantly between different treatments. We also identified some proteins related to solutes and oxidative enzymes by proteomic analysis, but these proteins were not significantly enriched in GO and KEGG analysis. Therefore, we did not discuss its pathway in this paper. However, the reviewer’s suggestion is very important. We will pay attention to this aspect in the future study.

(3) Line 464: Please improve the quality of figure 8.

Response: We used a clearer picture to replace the original one (Revision: Line 478).

Reviewer 2 Report

Dear editor

Please find enclosed my revision of the manuscript.

Manuscript Number: horticulturae-1422842
Title:

Drought-responsive proteins and metabolic pathways involved in
24-epibrassionlide regulating drought tolerance of Rhododendron delavayi

The paper suffers from several drawbacks that may need a minor revision. In any case, I highly encourage the authors to carefully review point by point to clarify some issues and eventually improve the manuscript.

There some problems in this manuscript

  1. Please the title must be changed to be more association with the research content..... there are not any interaction treatments between drought and brassionlide
  2. Please indicate how maintain irrigation regimes all over the experimental period, additionally, the pot (closed or having a leaching hole) in my opinion is very small for five plants till yield
  3. Please indicate in more detail the method of irrigation to be more repeatable
  4. Please indicate why this concentration 1 mg/L EBR in specific or indicate the references
  5. Line 112-114 what about control and brassionloid

Yours truly

Author Response

Response to Reviewer 2 Comment

The paper suffers from several drawbacks that may need a minor revision. In any case, I highly encourage the authors to carefully review point by point to clarify some issues and eventually improve the manuscript.

Response: Thank you for review’s suggestion. We try our best to revise the manuscript carefully. The details were as follows:

Title:

(1) Line 2: The title must be changed to be more association with the research content..... there are not any interaction treatments between drought and brassionlide

Response: We changed the title as ‘Key proteins and metabolic pathways involved in 24-epibrasionlide improving drought tolerance of Rhododen-dron delavayi. Franch’ (Revision: Line 2-4).

Abstract:

Line 16: ‘However, the mechanisms of BRs regulating the drought resistance of R. delavayi are not understood.’ There is not any interaction effect so delete or rewrite this sentence.

Response: We deleted this sentence and some abbreviations as reviewer’s suggestion (Revision: Line 15).

Introduction:

(1) Line 38, 42: The authors are suggested to cite more recent articles instead of old references.

We have cited the more recent articles instead of old references (Revision: Line 37-38, Reference 1,2,3).

(2) Line 48: The reviewer deleted ‘…both antioxidant enzymes and non-enzymatic antioxidants can be acculated…’.

Response: We revised this sentence (Revision: Line 47).

(3) Line 50-51: ‘In agricultural practices, growth regulators are often used for improving plant resistance to drought’. Revised they interested with stress factor only.

Response: We deleted this sentence to make this paragraph more focused (Revision: Line 49).

  1. Material and methods:

(1) Line 87: Open or closed it is very important to know how you can maintain the drought level.

Response: Yes, it is an open plastic pot. We added some information of the pot we used (Revision: Line 90-92).

(2) Line 93: Please indicate why this concentration 1 mg/L EBR in specific or indicate the references

Response: In our previous study, 1 mg/L EBR was the best concentration to improving drought tolerance of R. delavayi. We indicated the reference to support it (Revision: Line 99).

(3) Line 96-100: Please indicate how maintain irrigation regimes all over the experimental period, additionally, the pot (closed or having a leaching hole) in my opinion is very small for five plants till yield.

Response: We added more details about the methodology of stress application (Line 93-104). The pot we used is an open pot with a leaching hole, and one plant per pot (Revision: Line 86-87).

(4) Line 96-100: Please indicate in more detail the method of irrigation to be more repeatable.

Response: We added more details about the methodology of irrigation (Revision: Line 103-110).

(5) Line 112-114: What about control and brassionloid.

Response: We revised the sentence and added some information about the CK and BR treatment (Revision: Line 124-125).

Results:

Line 174-175: Figure 1, the figures is not well shown please colour it or changed to bar diagram.

Response: We redrawn Fig.1 with color (Revision: Line 185-186).

Discussion:

Line 282-283: ‘…In the present study, exogenous BR application improved photosynthetic performance of R. delavayi under drought conditions…’. The reviewer pointed out ‘there is no any interaction treatments.’

Response: Because we didn't clearly describe the method of different treatments in original manuscript, it may have caused some misunderstanding to the reviewers and felt that there were no any interaction treatments. The plants of BR group were pre-treated with BR, and then subject to the same drought level with the plants of D group. We added some details about its (Revision: Line 98-110). So, we believe that the treatment of BR group has interaction with drought, and the results also showed that exogenous BR application improved photosynthetic performance of R. delavayi under drought conditions.

  In addition, we also revised some minor mistakes pointed out by the reviewers. Please see the revised manuscript.

Reviewer 3 Report

The manuscript on Rhododendron delavayi under drought tolerance is significant work which can help this flowering plant to improve its yield at field level. I have major concerns about the manuscript which should be incorporated by authours.

Authours have not mentioned anywhere in the manuscript why this flowering plant was taken for the study? Is this prone to droght stress in the areas where it is cultivated if yes a detailed information must be given in the introduction section.

Authors have not also mentioned how drought treatment was given. Since it is a pot culture whether water was with-held during irrigation of PEG treatment was given to induce drought stress

Author Response

Response to Reviewer 3 Comment

(1) The manuscript on Rhododendron delavayi under drought tolerance is significant work which can help this flowering plant to improve its yield at field level. I have major concerns about the manuscript which should be incorporated by authours.

Authours have not mentioned anywhere in the manuscript why this flowering plant was taken for the study? Is this prone to droght stress in the areas where it is cultivated if yes a detailed information must be given in the introduction section.

Response: We revised and added a detailed information why we chose Rhododendron delavayi for our study. Please see the revised manuscript (Revision: Line 63-69).

(2) Authors have not also mentioned how drought treatment was given. Since it is a pot culture whether water was with-held during irrigation of PEG treatment was given to induce drought stress

Response: We revised and added more details how drought treatment was given. Please see the revised manuscript (Revision: Line 103-110).

Reviewer 4 Report

This article on drought stress amelioration through the exogenous application of 24-epibrassionlide hormone is an interesting manuscript and is a hot topic now a days due to climate change and water scarcity (the emerging issues of the century). Moreover, change in the proteomics of plant under different treatments is helpful in elaborating the drought tolerance mechanism of Rhododendron delavayi. The manuscript is written well with minimal grammatical errors and use state of the art techniques; therefore, I recommend publishing the manuscript for the sake of valuable contribution to the science with minor revision considering following suggestions in view:

Abstract: Organized and well written.

Introduction: Needs minor grammatical correction and rephrasing as described in the file attached.

Materials and methods: Need to clarify the treatments and include references of the protocols as described in the attached file.

Results: organized, well elaborated and clear to understand.

Discussion: Most of the portion consists of results repetitions with reasoning without references. Spike strong reasoning of the behavior of plant under the treatments with references.

Author Response

Response to Reviewer 4 Comment

This article on drought stress amelioration through the exogenous application of 24-epibrassionlide hormone is an interesting manuscript and is a hot topic now a days due to climate change and water scarcity (the emerging issues of the century). Moreover, change in the proteomics of plant under different treatments is helpful in elaborating the drought tolerance mechanism of Rhododendron delavayi. The manuscript is written well with minimal grammatical errors and use state of the art techniques; therefore, I recommend publishing the manuscript for the sake of valuable contribution to the science with minor revision considering following suggestions in view:

Abstract: Organized and well written.

Introduction: 

(1) Needs minor grammatical correction and rephrasing as described in the file attached.

Response: We have corrected all the grammatical errors as the reviewer’s suggestion. Please see the revised manuscript (Revision: Line 49, 52, 58, 63-69).

(1) Line 80: Clearly state the hypothesis of the study.

Response: We added the hypothesis of the study (Revision: Line 81-84) .

Materials and methods:

(1) Line 92-98: Need to clarify the treatments and include references of the protocols as described in the attached file.

Response: As the reviewer’s point out the description of this section is not clear. We rewrite this section to make its clearer (Revision: Line 90-110).

(2) Line 116, 134, 144: Methods need references to justify the prevailing conditions.

Response: We added a reference that uses the same method as we did (Revision: Line 130, Reference 20).

(2) Line 155: What was the statistical design used for the study.

Response: We used one-way ANOVA and LSD multiple comparisons tests to compared the effects of treatment (CK, BR, and D) on leaf physiological variables (Revision: Line 174-175).

Results: organized, well elaborated and clear to understand.

Discussion: 

Line 291: Most of the portion consists of results repetitions with reasoning without references. Spike strong reasoning of the behavior of plant under the treatments with references.

Response: We deleted some unnecessary duplicate results, and revised the line 304-308, and line 311-319. Also, we added some references to support our opinion (Revision: Reference 6, 7, 26, 28).

Reviewer 5 Report

This manuscript describes the key responsive proteins and metabolic pathways in the response of adaptation of Rhododendron delavayi in drought stress. This manuscript is valuable for all readers who study on the breeding of drought-tolerant rhododendrons.

There are several problems that should be corrected in the revised version. 

  1. Line 60-61:

Authors should give references to this sentence.

  1. Authors described the protein expression data in R. delavayi under drought stress with or without a BR pre-treatment. However, the reviewer wonders if the authors have validated protein abundance changes by Western blotting or not. Authors should confirm this works.

Finally, this manuscript will be accepted after major revision.

Author Response

Response to Reviewer 5 Comment

This manuscript describes the key responsive proteins and metabolic pathways in the response of adaptation of Rhododendron delavayi in drought stress. This manuscript is valuable for all readers who study on the breeding of drought-tolerant rhododendrons.

There are several problems that should be corrected in the revised version. 

(1) Line 60-61: Authors should give references to this sentence.

Response: We added reference to this sentence (Revision: Line 59, Reference 15).

(2) Authors described the protein expression data in R. delavayi under drought stress with or without a BR pre-treatment. However, the reviewer wonders if the authors have validated protein abundance changes by Western blotting or not. Authors should confirm this works.

Response: In present study, we employed a data independent acquisition (DIA) proteomics approach to describe the protein expression profiles in R. delavayi under drought stress with or without a BR pre-treatment. From our study, we found the key proteins and metabolic pathways involved in BR improving drought tolerance of R. delavayi. The article structure and main content focus on its. However, in present study, we didn’t validate protein abundance changes by Western blotting. As the reviewer’s suggestion, this work is very important for our further study, so we will validate the protein abundance changes in our future work. Thank you for reviewer’s suggestion again.

Round 2

Reviewer 5 Report

The authors revised the manuscript which was better than the previous one.

The reviewer gave the authors the agreement for publication in MDPI although the study was not interesting to the reader. 

Author Response

Thanks to the reviewer‘s  help and affirmation.  The reviewer gave us a lot of suggestions, which help us to revise and improve manuscript. In our future research, we will also improve the research program according to the opinions of the reviewers. Thank you again.